# Investigation of the Effect of Nutritional Supplementation with Whey Protein and Vitamin D on Muscle Mass and Muscle Quality in Subacute Post-Stroke Rehabilitation Patients: A Randomized, Single-Blinded, Placebo-Controlled Trial

**DOI:** 10.3390/nu14030685

**Published:** 2022-02-06

**Authors:** Kaoru Honaga, Naoki Mori, Tomonori Akimoto, Masahiro Tsujikawa, Michiyuki Kawakami, Tomoyuki Okamoto, Yasuyuki Sakata, Hirokazu Hamano, Yasuhiro Takeda, Kunitsugu Kondo

**Affiliations:** 1Department of Rehabilitation Medicine, Tokyo Bay Rehabilitation Hospital, 4-1-1 Yatsu, Narashino City 275-0026, Japan; hasilidoro@gmail.com (T.A.); tsujikawa-ma@hotmail.co.jp (M.T.); michiyukikawakami@hotmail.com (M.K.); muscle.kon@hotmail.com (K.K.); 2Department of Rehabilitation Medicine, Juntendo University Graduate School of Medicine, 2-1-1 Hongo, Bunkyo-ku, Tokyo 113-8421, Japan; 3Department of Rehabilitation Medicine, Hatsudai Rehabilitation Hospital, 3-53-3 Honmachi, Shibuya-ku, Tokyo 151-0071, Japan; vivanaoki@hotmail.com; 4Department of Rehabilitation Medicine, Keio University School of Medicine, 35 Shinanomachi, Shinjyuku-ku, Tokyo 160-8582, Japan; 5R & D Division, Morinaga Milk Industry Co., Ltd., 5-1-83 Higashihara, Zama City 252-8583, Japan; to-okamoto@morinagamilk.co.jp (T.O.); ys-sakata@morinagamilk.co.jp (Y.S.); h_hamano@morinagamilk.co.jp (H.H.); ya_taked@morinagamilk.co.jp (Y.T.)

**Keywords:** stroke, sarcopenia, rehabilitation, whey protein, vitamin D

## Abstract

In post-stroke hemiparesis patients, the skeletal muscle mass decrease rapidly with the histological degradation. We investigated the effect of nutritional supplementation with whey protein and vitamin D on the muscle mass and muscle quality, in post-stroke convalescent rehabilitation patients in a randomized, single-blinded, placebo-controlled trial. Fifty patients were randomly assigned to two groups; HP group received supplemental jelly (100 kcal; whey protein 10 g; vitamin D 20 μg) twice a day throughout up to 16-week period, the control group received placebo jelly. Cross-sectional area (CSA) of thigh muscle, skeletal muscle index (SMI), muscle strength, activity of daily living (ADL), and some nutritional indicators in blood were measured. Although no significant difference was observed in CSA and SMI between the groups, fat infiltration into the thighs muscle was singnificantly lower in the HP group. There were no significant difference in muscle strength and ADL between the groups. Blood urea nitrogen and serum 25(OH)D at endpoint were significantly higher in the HP group but physiological normal ranges. Supplementation with whey protein and vitamin D in post-stroke patients led to suppression of fat infiltration into the muscle. Long-term follow-up studies are needed to verify whether this nutritional intervention provides substantial benefits for the prognosis of stroke survivors.

## 1. Introduction

Stroke is the leading cause of long-term disability; one-third to two-thirds of survivors have hemiparesis [1]. In stroke patients with hemiparesis, muscle mass and strength rapidly decrease by progressive muscular atrophy due to neurological disorders, local inflammation, and immobility [2,3]. Moreover, as well as muscle mass and strength decrease, muscle quality is reduced histologically, such as a shift in muscle fiber type and intramuscular fat infiltration [3,4].

Sarcopenia is primarily defined as age-related loss of skeletal muscle mass and strength, and involves the qualitative deterioration of muscle tissue [5,6]. Sarcopenia is closely associated with poor quality of life and harmful outcomes in older people [7]; therefore, it is important to take countermeasures against sarcopenia. Physical exercise and good nutrition, especially adequate protein intake, are considered useful in preventing and improving this pathological muscle condition [8]. It is reported that bodyweight resistance exercise and supplementation with whey protein and vitamin D synergistically increase muscle mass in sarcopenic older adults [9].

Although age-related muscle loss is the principal etiological factor of sarcopenia, immobility, inflammation, insulin resistance, and undernourishment are also involved [5,10], that defined as secondary sarcopenia. The incidence of stroke increases with aging. Disability and reduced muscle function observed in stroke patients also contribute to the development of sarcopenia, which increases the risk of falls, insulin resistance, and mortality in the elderly [10,11,12,13]. Therefore, preventing stroke-related sarcopenia is of great interest not only for short-term rehabilitation gain but also for long-term outcomes in stroke survivors.

Among old age, upper limb dysmobility, dysphagia, and others, post-stroke patients often suffer the risk of nutritional deterioration [14], and several studies have suggested the possibility that nutritional intervention improves rehabilitation outcomes and physical function in such patients [15]. Yoshimura et al. reported that leucine-enriched amino acid supplementation and resistance training in post-stroke sarcopenic patients increased muscle mass, strength, and physical function [16]. Regarding the effects of nutritional supplementation combined with exercise, not only specific amino acid mixture, but also dietary proteins, including cow’s milk whey protein, are known to improve muscle mass and function in the elderly [17,18]. Furthermore, it has been suggested that vitamin D supplementation improves muscle metabolism and has beneficial effects, especially in elderly people [19,20,21]. It has been reported that nutritional supplements containing whey protein and vitamin D combined with exercise enhance the muscle mass in mobility-limited older adults, including inpatients [22,23]. However, there are limited studies on the effects of supplementation with dietary protein and vitamin D on outcomes in post-stroke rehabilitation patients.

In this study, we investigated the effect of nutritional supplementation with whey protein and vitamin D on the outcomes, particularly muscle mass and muscle quality (muscle attenuation), in post-stroke convalescent rehabilitation patients in a randomized, single-blind, placebo-controlled trial.

## 2. Materials and Methods

### 2.1. Participants

Consecutive patients with ischemic or hemorrhagic stroke admitted to Tokyo Bay Rehabilitation Hospital, Chiba, Japan, were screened for this study. Fifty patients who met the following inclusion criteria were enrolled: age ≥ 40 years with hospital admission ≤ 2 weeks, with expected hospitalization period ≥ 2 months, with functional independence measure (FIM) walk/wheelchair ≤ 4, social interaction ≥ 6, comprehension ≥ 5, and expression ≥ 5, with the ability to use a normal wheelchair, were enrolled. The exclusion criteria were as follows: the presence of cognitive deficits (thereby making informed consent obtainment impossible), contraindications to exercise therapy due to complications or previous diseases, difficulty swallowing supplements, receiving parenteral or enteral nutrition, an estimated glomerular filtration rate < 60 mL/min/1.73 m^2^, regular use of protein or vitamin D or citrus peel supplements, and serious allergy to foods or medicines.

This study was approved and monitored by the Institutional Review Board of Tokyo Bay Rehabilitation Hospital (No. 121) and followed the principles of the Declaration of Helsinki. The study protocol was registered at the UMIN Clinical Trial Registry (UMIN ID: 000019360) before the first participant was included. All participants provided written informed consent before participating in the study.

### 2.2. Study Design

This study was a single-blinded, placebo-controlled, randomized trial. Participants were randomly assigned to the HP group or the control group using a permuted block design with a computer random number generator. The block size was four, and stratification according to sex was applied. Random allocation was performed by a person who was independent of the study. The randomization code was opened after all data were checked, collated, and finalized.

### 2.3. Intervention and Materials

The subjects received conventional stroke rehabilitation in the convalescent rehabilitation ward. Subjects in the HP group were supplied with high-protein jelly type supplement (100 kcal; 10 g of whey protein; 20 μg of vitamin D, *Riha-Time* Jelly, CLINICO, Tokyo, Japan) twice a day throughout the 16-week study period. Study period was set considering the previous duration of hospitalization of the patients met the criteria in this study in our hospital. The control group received placebo jelly (32 kcal; 0 g of protein; 0 μg of vitamin D). The subjects were blinded to group allocation. The daily intake of supplementary jelly and hospital meals was recorded by nursing staff every day. Hospital meal was individualized to implement each energy requirement calculated by Harris-Benedict equation [24] using ideal body weight based on body mass index (BMI) of 22.0 with stress factor 1.0 and activity factor 1.3. Protein content in the hospital meal was set at 1.0 g/kg ideal body weight/day. Amount of protein supplementation in the HP group was designed as sufficient amount for a 1.2–1.5 g/kg body weight recommended by the ESPEN Expert Group for the elderly having acute or chronic illness in total with hospital meal [8].

### 2.4. Outcome Measures

The primary outcome measures were skeletal muscle index (SMI) and cross-sectional area (CSA) of the thigh muscles. SMI was defined as appendicular muscle mass, assessed by bioelectrical impedance analysis (Physion MD, Nippon Shooter Ltd., Tokyo, Japan), divided by height squared (kg/m^2^). CSA was measured using a six-detector row CT scanner (Brilliance 6; Royal Philips, Amsterdam, Netherlands) at 20 cm above the upper edge of the patella to quantify the skeletal muscle area of the paretic and non-paretic thighs. Six slices of the thigh image were obtained by one rotation of the CT scan. The slice interval was 1.5 mm. The obtained data were analyzed using the OsiriX 8.0 viewer (OsiriX Foundation, Geneva, Switzerland). The thigh muscle CSA was quantified in the range from −30 to 150 Hounsfield units (HU). In addition, the thigh muscle CSA was partitioned into normal-density muscle CSA (30 to150 HU) and low-density muscle CSA (−30 to 29 HU), defined as the muscle area with fat infiltration [25].

The secondary outcome measures were muscle strength, activities of daily living (ADL), physical function, length of hospital stay, and some nutritional indices (serum albumin, pre-albumin, insulin-like growth factor (IGF)-1, and 25(OH)D). For muscle strength, we assessed hand grip strength and lower extremity strength. The isometric knee extensor strength at 60° knee flexion was measured using a hand-held dynamometer (JTech Power Track II; JTech Medical, Salt Lake City, UT, USA). The leg extensor torque was measured using a position-controllable cycle ergometer (StrengthErgo; Mitsubishi Electric Co., Tokyo, Japan).

For the assessment of ADL, the motor domain of the functional independence measure (FIM-M) was used [26]. The FIM-M contains 13 ADL-related items, and each item is scored from 1 to 7 (total score = 91). The 10-m comfortable and maximum walk test [27], time up and go test [28], and 30-sec chair stand test [29] were performed to assess physical function.

Moreover, we performed exhaled breath analysis to measure the resting energy expenditure (REE) of the subjects. A portable exhaled gas analyzer (Aerosonic AT-1100A; Anima Co., Tokyo, Japan) was used.

For the safety analysis, adverse events, physical findings, and blood examinations (hematological tests, and basic serum biochemical testes such as hemoglobin A1c (HbA1c), triglyceride, total cholesterol, blood urea nitrogen (BUN)) were performed in all registered participants.

Measurements were performed at baseline and at the end of the 16-week-intervention period. If the subject was discharged before 16 weeks, the intervention was completed, and measurements were performed at that time. The time of the final measurement was defined as the study endpoint.

### 2.5. Statistical Analysis

The demographic characteristics, baseline clinical data, and amount of physical therapy were compared between the two randomized groups using the unpaired *t*-test and Fisher’s exact test for continuous and categorical variables, respectively. Supplemental jelly consumption in both groups was compared using the Wilcoxon rank-sum test. Intergroup comparisons were performed using covariance analysis with covariate value at 0 week as allocation factor. The change in outcome measures from the baseline values was analyzed using the paired *t*-test. Statistical analyses were performed using JMP version 13.0 (SAS Institute, Cary, NC, USA). Statistical significance was set at *p <* 0.05.

## 3. Results

### 3.1. Study Participants

From November 2015 to November 2018, a total of 50 participants who met the inclusion criteria were equally randomized into the HP and placebo groups. Three participants from the HP group (one for missing data during the intervention period, two for protocol violation) and two participants from the control group (both for protocol violation) were excluded, resulting in 45 participants (22 in the HP group and 23 in the placebo group) included in the analysis of the primary and secondary outcome measures (Figure 1).

There was no significant difference in the demographic characteristics between the groups (Table 1). This study included more patients with hemorrhagic stroke compared with ischemic, as a result of the exclusion by cognitive deficits of some patients with ischemic stroke. The consumption rates of hospital meals and supplemental jellies and the amount of physical therapy are shown in Table 2. Although the consumption rate of hospital meals was somewhat low in the HP group, there was no statistical difference in these parameters.

### 3.2. Primary Outcomes

The primary outcomes, SMI, and CSA, are shown in Table 3. No significant difference in SMI was observed between the HP and placebo groups at the end of the intervention period. Similarly, CSA in both paretic and non-paretic thighs did not show significant differences between the groups. However, CSA with fat infiltration, defined by low CT density ranging from −30 to 29 HU, was significantly lower in the paretic and non-paretic thighs in the HP group.

### 3.3. Secondary Outcomes and Others

Secondary outcome measures are presented in Table 4. Muscle strength, FIM-M, and physical function were improved significantly during intervention in both groups, however, no significant difference between the groups. Meanwhile, serum 25(OH)D and REE levels were significantly higher in the HP group. The length of hospital stay was not different between the groups: 117.8 ± 3.4 and 107.7 ± 4.2 days in the HP and control groups, respectively (*p* = 0.217). Blood urea nitrogen (BUN) was significantly higher in the HP group, whereas these were in the normal range. Although the ratio of HbA1c in the HP group was significantly higher than that in the control group at the endpoint, there was no change from baseline (Table 5). There were not observed significant difference in BMI and triglyceride (TG) between the groups, whereas significant decreases were observed from baseline only in the HP group (both *p* < 0.05). No adverse events related to supplemental jellies were observed.

## 4. Discussion

In this study, nutritional supplementation with whey protein and vitamin D in post-stroke rehabilitation patients had no positive impact on muscle mass and physical function. The nutritional intervention in this study was the administration of supplemental jelly containing 10 g of whey protein and 20 μg of vitamin D, or placebo jelly, which was administered in two portions per day. Cow’s milk whey protein is a good quality dietary protein with high digestibility and is rich in indispensable amino acids. There are many reports that whey protein supplementation combined with exercise increases muscle protein synthesis in both young and elderly subjects [30,31,32,33]. Likewise, vitamin D has been suggested to affect skeletal muscle function [19], and its supplementation is recommended, especially for elderly individuals with sarcopenia or frailty [20,21]. Indeed, it has been reported that the combined intervention of whey protein and vitamin D supplementation and body weight resistance exercise increases skeletal muscle mass in community-dwelling older adults with sarcopenia [9]. Furthermore, a previous study showed that the ingestion of whey protein and vitamin D enriched jelly type supplement, similar to that used in this study, increased skeletal muscle mass in rehabilitation patients, including stroke survivors [23].

In contrast, in this study, we did not observe significant effects of nutritional supplementation of whey protein and vitamin D on muscle mass, strength, and physical function. The factors that may influence nutritional intervention outcomes are the dose of the supplement, the level of combined exercise, and the attribution of subjects. In this study, significant increases in serum BUN and 25(OH)D were observed in the HP group subjects (0 week vs. endpoint, both *p* < 0.01). Therefore, the supplemented dose of whey protein and vitamin D was probably sufficient to exert physiological effects in the subjects. However, it has been suggested that vitamin D status is important in order to acquire maximum extraskeletal actions of vitamin D including adipose/muscle cells transdifferentiation, and serum 25(OH)D levels above 40 ng/mL would be ideal [34]. Further study might be needed to evaluate the effective dose of vitamin D in post-stroke patients. Regarding the exercise level and subjects’ attribution, there might have been a substantive difference between the exercise regimen for the rehabilitation of post-stroke hemiparetic patients and bodyweight resistance exercise in healthy seniors. Although vitamin D supplementation has been shown to increase muscle strength in sarcopenic elderly, Momosaki et al. reported that vitamin D supplementation did not improve muscle strength and ADL in post-stroke rehabilitation patients [35]. Physical intervention to the subjects in this study was gait training or occupational therapy as post-stroke rehabilitation, so protein and vitamin D supplementation might have been difficult to contribute the muscle mass or muscle strength in these subjects. Regarding the attribution of subjects, in the previous study using the same nutritional supplement, the majority (from 3/4 to 4/5) of subjects had underlying diseases other than stroke, and nutritional intervention in these subjects started from the acute phase [23]. In the post-stroke patients with gait impairment (FIM ≤ 4 in this study), hardly perform to sufficient resistance exercise, whey protein and vitamin D supplementation might have limited effectiveness on the muscle mass.

On the other hand, fat infiltration into the muscle was suppressed in the HP group. In addition, significant decrease in TG and BMI from baseline were observed only in the HP group, meanwhile SMI were maintained from baseline. It was implied that the amount of meal calculated by the Harris-Benedict equation was enough to maintain body weight, because decreasing BMI from baseline was not observed in the Placebo group. Therefore, these changes in the HP group indicate a decrease in body fat including fat infiltration into the muscle, while maintaining mucle mass. In addition, in this study, significant increase in REE was observed in the HP group compared with Placebo group. It has been reported that whey protein ingestion induces to decrease body weight and body fat [36]. Furthermore, whey protein ingestion has also induced to increase brown adipose tissue which affects energy expenditure and to regulate muscle lipid and fatty acid metabolism by decreasing the mRNA levels in the animal studies [37,38].

Similarly, vitamin D deficiency induces to increase body fat and fat infiltration into the muscle and to inhibit muscle protein synthesis in the aged animal [39]. Vitamin D insufficiency is commonly observed in the patient admitted in the hospital, including the patient ingesting vitamin D of the recommended daily amount [40]. Furthermore, lower 25(OH)D level are known to associate with the risk of stroke onset [41]. We also observed lower serum 25(OH)D in our patients. However in this study, serum 25(OH)D level increased to normal range during the intervention period in the HP group. Vitamin D supplementation with 40 μg (1600 IU)/day was sufficient to improve the serum 25(OH)D level in post-stroke rehabilitation patients. Benefical effect of vitamin D supplementation on serum fat profile such as TG was also reported in the human study [42]. Therefore, improvement of BMI, serum TG and fat infiltration into the muscle might have been contributed by the both of whey protein and vitamin D supplementation in this study.

Regarding fat infiltration into the muscle, in post-stroke hemiparetic patients, not only a decrease in skeletal muscle mass, but also an increase in fat deposition is observed in both the surrounding and inside of the muscle [4]. Intramuscular fat accumulation indicates as the low-density lean tissue in CT, that is, increased muscle attenuation. It has been reported that muscle attenuation in post-stroke hemiparetic patients is negatively associated with fasting plasma insulin levels, which means that the increased intramuscular fat causes insulin resistance and relates to the risk of type 2 diabetes [43]. Therefore, suppressing fat infiltration into the skeletal muscle would reduce the risk of complications such as diabetes and contribute to the improvement of prolonged outcomes in stroke survivors. Although HbA1c level in the HP group was not decreased, this is probably because the average HbA1c at 0 week was within the normal range.

Intramuscular fat accumulation is also increases with aging [44]. Increased intramuscular fat accumulation is associate with muscle strength [45] and is known to be a risk of mobility restriction [46]. In addition, reduced muscle quality has been included into the operational definition of sarcopenia on EWGSOP2 [5]. Therefore, decreased intramuscular fat infiltration observed in this study may suppress the progression of sarcopenia in post-stroke elderly patients by maintaining their muscle quality

There are several limitations in this study. First, we have no detailed information about the nutritional intakes from the daily meals. The subjects in this study consumed hospital meal at the same facility. The amount of energy and protein in the hospital meal were set based on ideal body weight individually. It is considered that there were no large differences in the macronutrients between the groups. However, the amounts of micronutrients could not identify, these differences might have been affected the results. Second, we used commercially available products to evaluate the effect of whey protein and vitamin D on the muscle. This study design could not allow us to evaluate the contribution of each nutrients individually. Further study will be needed on each nutrients individually.

## 5. Conclusions

Although no significant change was observed in muscle mass and strength, nutritional intervention with whey protein and vitamin D in patients with subacute post-stroke hemiparesis led to suppression of fat infiltration into the muscle. Long-term follow-up studies are needed to verify whether this nutritional intervention provides substantial benefits for the prognosis of stroke survivors.

## Figures and Tables

**Figure 1 nutrients-14-00685-f001:**
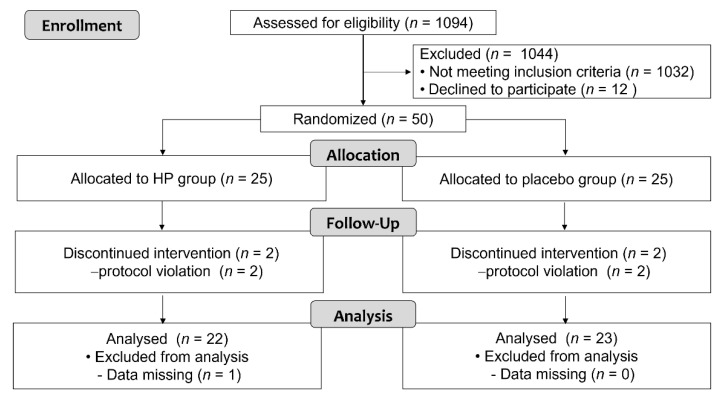
Participant flow.

**Table 1 nutrients-14-00685-t001:** Demographic characteristics of the study participants.

	HP Group	Placebo Group	*p* Value
Age (years)	64.2 ± 8.9	61.3 ± 11.5	0.354
Gender (Male/Female)	15/7	15/8	0.833
Bodyweight (kg)	61.8 ± 10.3	58.9 ± 12.4	0.405
BMI (kg/m^2^)	22.4 ± 2.8	22.1 ± 3.0	0.732
Types of stroke			
(Ischemic/Hemorrhagic)	9/16	9/16	1.000
Time from onset to rehabilitation admission			
(days)	29.0 ± 11.7	33.5 ± 12.1	0.192
Complications			
(No. of hypertension)	22	21	1.000
(No. of diabetes mellitus)	3	1	0.609
(No. of hyperlipidemia)	5	5	1.000

Excepted gender, types of stroke, and complications, values are presented as mean ± standard deviation. BMI, body mass index.

**Table 2 nutrients-14-00685-t002:** Dietary consumption rates and amounts of physical therapy during intervention periods.

	HP Group	Placebo Group	*p* Value
Dietary consumption rates			
Supplemental jellies (%)	96.3 ± 3.1	95.6 ± 4.1	0.751
Hospital meal (point)	9.0 ± 1.5	9.6 ± 0.6	0.089
Physical therapy (min./day)	70 ± 6	68 ± 8	0.694

Consumption rates of hospital meals were evaluated by the point on 0 to 10; Values are presented as mean ± standard deviation.

**Table 3 nutrients-14-00685-t003:** Changes in the primary outcome measures during the intervention periods.

	HP Group		Placebo Group	
	0 Week	Endpoint	0 Week	Endpoint
SMI (kg/m^2^)	4.70 ± 0.95 (22)	4.48 ± 0.58 (22)	4.65 ± 0.74 (23)	4.54 ± 0.66 (23)
CSA (cm^2^)				
Total muscle area				
Paretic side	87.0 ± 20.3 (22)	88.5 ± 20.5 (22)	90.3 ± 22.7 (23)	94.1 ± 27.1 (22)
Non-paretic side	99.8 ± 22.8 (22)	103.9 ± 20.7 (22)	104.8 ± 26.2 (23)	111.3 ± 26.7 (22)
Normal muscle area				
Paretic side	62.7 ± 16.0 (22)	65.7 ± 16.7 (22)	66.5 ± 17.2 (23)	68.8 ± 20.5 (22)
Non-paretic side	74.5 ± 17.6 (22)	81.1 ± 16.6 (22)	80.1 ± 19.4 (23)	85.3 ± 19.7 (22)
Muscle area with fat infiltration				
Paretic side	24.3 ± 7.0 (22)	22.8 ± 6.6 (22) ^#^	23.8 ± 7.7 (23)	25.3 ± 8.2 (22) ^#^
Non-paretic side	25.2 ± 7.7 (22)	22.9 ± 6.8 (22) ^#^	24.7 ± 8.8 (23)	26.0 ± 8.9 (22) ^#^

Values are presented as mean ± standard deviation (n); ^#^: significant difference between the groups, *p <* 0.05. SMI, skeletal muscle index; CSA, cross-sectional area.

**Table 4 nutrients-14-00685-t004:** Changes in the secondary outcome measures and during the intervention periods.

	HP Group		Placebo Group	
	0 Week	Endpoint	0 Week	Endpoint
Muscle strength				
Hand grip (kgf)				
Paretic side	4.7 ± 8.1 (22)	5.5 ± 7.8 (21)	5.4 ± 8.1 (22)	8.9 ± 9.6 (20)
Non-paretic side	27.8 ± 9.3 (22)	28.1 ± 8.3 (20)	26.2 ± 10.4 (22)	28.4 ± 9.8 (20)
Isometric knee extensor strength (Nm)				
Paretic side	28.2 ± 22.7 (22)	45.6 ± 37.9 (21)	29.3 ± 23.9 (20)	49.3 ± 40.8 (19)
Non-paretic side	49.3 ± 31.3 (22)	65.5 ± 42.0 (21)	53.0 ± 38.0 (20)	72.5 ± 43.8 (19)
Leg extensor torque (N)				
Paretic side	56.9 ± 63.5 (21)	90.3 ± 63.1 (22)	74.3 ± 72.3 (22)	134.3 ± 92.6 (19)
Non-paretic side	129.2 ± 59.6 (21)	150.3 ± 70.7 (22)	155.5 ± 81.3 (22)	192.8 ± 75.2 (19)
ADL				
FIM-Motor	49.2 ± 15.9 (22)	73.1 ± 14.2 (22)	49.8 ± 15.0 (23)	74.4 ± 18.3 (23)
Physical function				
10-m walk test (m/s)				
Comfortable	0.31 ± 0.28 (22)	0.60 ± 0.38 (21)	0.30 ± 0.30 (22)	0.69 ± 0.38 (21)
Maximum	0.38 ± 0.44 (22)	0.76 ± 0.53 (21)	0.39 ± 0.42 (21)	0.90 ± 0.55 (20)
Time Up and Go (s)				
Paretic side	102.5 ± 128.7 (22)	28.2 ± 27.0 (21)	98.7 ± 125.4 (21)	20.7 ± 15.2 (20)
Non-paretic side	106.6 ± 129.7 (22)	28.1 ± 26.6 (21)	99.7 ± 126.5 (21)	21.4 ± 16.8 (20)
30-s Chair test (*n*)	7.4 ± 4.4 (22)	11.2 ± 7.6 (22)	8.3 ± 4.2 (21)	13.5 ± 5.8 (18)
Nutritional indicators and others				
Alb (g/dL)	3.8 ± 0.3 (22)	4.0 ± 0.3 (22)	3.9 ± 0.3 (23)	4.1 ± 0.4 (23)
pre-Alb (mg/dL)	25.3 ± 6.6 (22)	25.6 ± 5.5 (22)	28.6 ± 7.3 (23)	29.3 ± 7.9 (22)
IGF-1 (ng/dL)	125.3 ± 44.5 (22)	131.2 ± 50.2 (22)	154.1 ± 39.8 (23)	147.5 ± 41.4 (22)
25(OH)D (ng/mL)	15.2 ± 5.1 (22)	25.2 ± 4.7 (22) ^#^	17.0 ± 7.0 (23)	17.2 ± 5.9 (23) ^#^
REE (kcal/kgBW/day)	22.5 ± 3.4 (22)	26.2 ± 4.6 (22) ^#^	23.5 ± 4.5 (23)	23.1 ± 3.3 (22) ^#^

Values are presented as mean ± standard deviation (n); ^#^: significant difference between the groups, *p* < 0.05; ADL, activities of daily living; FIM-Motor, motor domain of the Functional Independence Measure; Alb, albumin; IGF-1, insulin-like growth factor-1; REE, resting energy expenditure.

**Table 5 nutrients-14-00685-t005:** Changes in the physical findings and serum states.

	HP Group		Placebo Group	
	0 Week	Endpoint	0 Week	Endpoint
BMI (kg/m^2^)	22.4 ± 2.8 (25)	21.5 ± 2.3 (24)	21.9 ± 3.1 (25)	21.7 ± 3.0 (25)
Serum states				
TG (mg/dL)	110.6 ± 35.8 (25)	92.8 ± 28.0 (24)	124.7 ± 54.8 (25)	123.9 ± 85.9 (25)
HbA1c (%)	5.7 ± 0.6 (25)	5.6 ± 0.4 (24) ^#^	5.7 ± 0.4 (25)	5.4 ± 0.3 (25) ^#^
BUN (mg/dL)	11.7 ± 2.9 (25)	15.5 ± 3.9 (24) ^#^	11.1 ± 2.7 (25)	11.6 ± 3.0 (25) ^#^

Values are presented as mean ± standard deviation (n); ^#^: significant difference between the groups, *p* < 0.05; BMI, body mass index; TG, triglyceride; HbA1c, hemoglobin A1c; BUN, blood urea nitrogen.

## Data Availability

Original data is available with the corresponding author but not archived in database elsewhere.

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
