# Peer review of "Investigation of the Effect of Nutritional Supplementation with Whey Protein and Vitamin D on Muscle Mass and Muscle Quality in Subacute Post-Stroke Rehabilitation Patients: A Randomized, Single-Blinded, Placebo-Controlled Trial"

_nutrients, 2022, doi:10.3390/nu14030685_

Round 1

Reviewer 1 Report

The article may be interesting, as it deals with a nutritional supplement for people who have suffered a stroke.

Regarding the introduction, you have to enter a value that is important, such as the time elapsed until the hospital, I recommend this appointment:

Soto-Cámara R, González-Santos J, González-Berna J, Trejo-Gabriel-Galán JM. Factors associated with a rapid call for assistance for patients with ischemic stroke. Emergencias. 2020 Feb;32(1):33-39. English, Spanish. PMID: 31909910.

The sample has many hemorrhages, although ischemic stroke is more frequent, this must be explained.

Regarding the study design, the amount and time of treatment must be specified.

Regarding the analysis of results, the necessary test is an ancove of the differential score, having the pretest as a covariate. A comparison of the profit of each group is necessary.

Regarding the results, the few significant differences in some sections are striking, despite the gain, the ancova will be necessary and confirm this.

It is a study financed by an interested company, so the results must be much more reliable, that is, it is necessary that the conclusions be more precise, according to the discussion and results obtained.

Author Response

20 January 2022

Dear Reviewer

I wish in response to the decision on our manuscript, titled ‘Nutritional supplementation with whey protein and vitamin D in patients undergoing subacute post-stroke rehabilitation suppresses fat infiltration into the muscle: A randomized, single-blinded, placebo-controlled trial’ (Manuscript ID.: nutrients-1532250).

We are grateful for your comments regarding our manuscript. We have carefully revised the manuscript according to these suggestions, addressing each comment to the best of our ability.

Indeed, the suggestions were very insightful and useful in revising this manuscript. Along with the revised manuscript, please find attached a point-by-point reply to each of the reviewer comments.

Thank you for your continued consideration. We hope that our responses and revisions are satisfactory, and that the manuscript is now deemed to be suitable for publication in Nutrients. I look forward to hearing from you.

Reply to Reviewers' comments

The article may be interesting, as it deals with a nutritional supplement for people who have suffered a stroke.

Regarding the introduction, you have to enter a value that is important, such as the time elapsed until the hospital, I recommend this appointment:

Soto-Cámara R, González-Santos J, GonzálezBerna J, Trejo-Gabriel-Galán JM. Factors associated with a rapid call for assistance for patients with ischemic stroke. Emergencias.

2020 Feb;32(1):33-39. English, Spanish. PMID: 31909910.

Reply:

We greatly appreciate for reviewer for providing many important comments. We are thankful for the time and energy you expended.

Thank you for the comment on this important point and introducing interesting paper. Absolutely, the time from the onset to hospital is one of the most crucial factors for the treatments in stroke, especially acute phase. However, the purpose of this study was to evaluate the intensive nutritional therapy for stroke patient with rehabilitation in subacute and convalescent phase, not in acute phase. In Japanese health care system, stroke patients change hospital from acute care hospitals admitted at onset of stroke to convalescent rehabilitation facility within 60 days. The participants in this study were stroke patient of subacute to convalescent phase in the convalescent rehabilitation facility. We consider that it might not appropriate to mention this point in this manuscript, considering the difference of the stroke stage of the participants in this study.

The sample has many hemorrhages, although ischemic stroke is more frequent, this must be explained

Reply:

Thank you for the comment. As reviewer pointed out, ischemic stroke is more frequent than hemorrhagic especially in older age. In this study, many older patients with ischemic stroke were excluded by the exclusion criteria such as cognitive function, more patients with ischemic stroke were allocated as a result.

We have added the following sentence (line 180- 182).

“This study included more patients with hemorrhagic stroke compared with ischemic, as a result of the exclusion by cognitive deficits of some patients with ischemic stroke.                                                     “

Regarding the study design, the amount and time of treatment must be specified.

Reply:

Thank you for this important suggestion.

Regarding the amount of treatment in this study, it was set as “1.2 - 1.5 g protein intake / kg body weigh / day” which is recommended dosage of the ESPEN Expert Group for the elderly having acute or chronic illness. In our facility, patients are provided 1.0 g protein / ideal body weight / day through hospital meal. Supplementation with 20 g of whey protein (2 packs of high-protein jelly) on hospital meal meets the recommended amount.

We have added the following sentence (line 119 - 123).

“Protein content in the hospital meal was set at 1.0 g / kg ideal body weight / day. Amount of protein suppelemtation in the HP group was designed as sufficient amount for a 1.2 - 1.5 g / kg body weight recommended by the ESPEN Expert Group for the elderly having acute or chronic illness in total with hospital meal.”

Regarding the time of treatment, it was set as the longest duration prospected from previous patients who met criteria in this study.

Also, we have added the following sentence (line 112 - 114)

“Study period was set considering the previous duration of hospitalization of the patients met the criteria in this study in our hospital.”

Regarding the analysis of results, the necessary test is an ancove of the differential score, having the pretest as a covariate. A comparison of the profit of each group is necessary.

Reply:

Thank you for the comment, we agree with the reviewer’s opinion about statistics. We used baseline data (0 week) as covariate values and did not perform intergroup analysis at baseline.

To clarify, we have changed the following sentence from (Line 165 - 168)

“Intergroup comparisons were performed using covariance analysis.”

To

“Intergroup comparisons were performed using covariance analysis with covariate value at 0 week as allocation factor.”

Regarding the results, the few significant differences in some sections are striking, despite the gain, the ancova will be necessary and confirm this.

Reply:

Thank you for the comment. As mentioned above, statistical analysis was performed at all outcomes described in our paper appropriately. Actually, although BMI and TG decreased significantly from baseline to endpoint in HP group (p < 0.05), there were no significant differences between the groups at endpoint (p = 0.157 and p = 265, respectively).

To clarify, we have added following sentence (Line 219 - 221)

“There were not observed significant difference in BMI and triglyceride (TG) between the groups, whereas significant decreases were observed from baseline only in the HP group (both p < 0.05).”

It is a study financed by an interested company, so the results must be much more reliable, that is, it is necessary that the conclusions be more precise, according to the discussion and results obtained.

Reply:

We greatly appreciate for reviewer for raising this important point. We have considered that it should not to be mentioned directly about the risk of metabolic complications in conclusion.

We have changed the following sentence in Abstract from (Line 33 - 37)

“Supplementation with whey protein and vitamin D in post-stroke patients led to suppression of fat infiltration into the muscle, which was expected to reduce the future risk of metabolic complications. Long-term follow-up studies are needed.”

To

“Supplementation with whey protein and vitamin D in post-stroke rehabilitation patients led to suppression of fat infiltration into the muscle. Long-term follow-up studies are needed to verify whether this nutritional intervention provides substantial benefits for the prognosis of stroke survivors.”

Also, we have changed the following sentence in Conclusion section from (Line 366 - 371)

“Nutritional intervention with whey protein and vitamin D in patients with subacute post-stroke hemiparesis led to suppression of fat infiltration into the muscle. Increased intramuscular fat induces insulin resistance, and suppressing intramuscular fat accumulation is expected to reduce the risk of metabolic complications in stroke survivors.”

To

“Although no significant change was observed in muscle mass and strength, Nutritional intervention with whey protein and vitamin D in patients with subacute post-stroke hemiparesis led to suppression of fat infiltration into the muscle.”

Again, thank you for giving us the opportunity to strengthen our manuscript with your valuable comments and queries. We have worked hard to incorporate your feedback and hope that these revisions persuade you to accept our submission.

Sincerely,

Kaoru Honaga

Department of Rehabilitation Medicine, Juntendo University Graduate School of Medicine

2-1-1, Hongo, Bunkyo Wardy, Tokyo 113-8421, Japan

Tel:  +81-3-3813-3111, Fax: +81-3-5684-1861,

E-mail address: k.honaga.ps@juntendo.ac.jp

Reviewer 2 Report

The objective of this RCT is to evaluate the effect of nutritional supplementation with whey protein and vitamin D on muscle mass and quality in post-stroke rehabilitation patients.

The authors report data from 2015-2018 and are proposed for publication 4 years after completion of the work.

There are some methodological shortcomings that prevent publication in this form.

  1. Most outcomes were not shown to be statistically significant between supplemented and control groups. The authors should emphasize the limited usefulness of supplementation for improving the nutritional status of subjects.
  2. Supplementing with vitamin D and wheyprotein combined does not allow us to understand which of the two supplements may have given the improvement in intra-muscular fat observed in the study group.
  3. Many nutritional data have not been evaluated. Dosage of pre-albumin, tranferrin, and complement factors would have been very useful in assessing whether supplementation may have had real utility in improving the body composition status of the subjects. 
  4.  There are important differences between the two groups after the supplementation period. The study group had a major reduction in weight and TGs. How do the authors justify these differences with the placebo group?
  5.  It would be useful to evaluate the vitamin D blood data from the beginning of supplementation and the end.
  6. Dietary assessment is almost absent. Knowing the type of diet followed and any differences in the quality and quantity of intake of macronutrients is essential.
  7. The discussion in which the possible benefit of intramuscular fat reduction on the risk of insulin resistance and diabetes is emphasized seems very forced and with little justification. The authors should revise the discussion and also emphasize the poor effects of supplementation considering also the costs.

Author Response

20 January 2022

Dear Reviewer

I wish in response to the decision on our manuscript, titled ‘Nutritional supplementation with whey protein and vitamin D in patients undergoing subacute post-stroke rehabilitation suppresses fat infiltration into the muscle: A randomized, single-blinded, placebo-controlled trial’ (Manuscript ID.: nutrients-1532250).

We are grateful for your comments regarding our manuscript. We have carefully revised the manuscript according to these suggestions, addressing each comment to the best of our ability.

Indeed, the suggestions were very insightful and useful in revising this manuscript. Along with the revised manuscript, please find attached a point-by-point reply to each of the reviewer comments.

Thank you for your continued consideration. We hope that our responses and revisions are satisfactory, and that the manuscript is now deemed to be suitable for publication in Nutrients. I look forward to hearing from you.

Reply to Reviewers' comments

The objective of this RCT is to evaluate the effect of nutritional supplementation with whey protein and vitamin D on muscle mass and quality in post-stroke rehabilitation patients.

The authors report data from 2015-2018 and are proposed for publication 4 years after completion of the work.

There are some methodological shortcomings that prevent publication in this form.

Reply:

We greatly appreciate for reviewer for providing many important comments and useful suggestions. We are thankful for the time and energy you expended.

We believe that our data do not include inappropriate procedure.

  1. Most outcomes were not shown to be statistically significant between supplemented and control groups. The authors should emphasize the limited usefulness of supplementation for improving the nutritional status of subjects.

Reply:

Thank you for the comment on this important point. As reviewer pointed out, our nutritional intervention could not improve muscle mass, strength, and physical function in post-stroke rehabilitation patients. We speculate that more strong resistance exercise was required to gain muscle mass and strength with nutrition intervention.

In Discussion section, we have separated the paragraphs of the argument that there was no effect on muscle mass and strength and the argument that intramuscular fat was reduced, emphasized the former. (Line 233 - 270)

Also, we have added following sentence (Line 268- 270)

“In the post-stroke patients with gait impairment (FMI ≤ 4 in this study), hardly perform to sufficient resistance exercise, whey protein and vitamin D supplementation might have limited effectiveness on the muscle mass.”

Also, we have changed the following sentence in Abstract from (Line 33 - 37)

”Supplementation with whey protein and vitamin D in post-stroke patients led to suppression of fat infiltration into the muscle, which was expected to reduce the future risk of metabolic complications. Long-term follow-up studies are needed.”

To

“Supplementation with whey protein and vitamin D in post-stroke rehabilitation patients led to suppression of fat infiltration into the muscle. Long-term follow-up studies are needed to verify whether this nutritional intervention provides substantial benefits for the prognosis of stroke survivors.”

Also, we have changed the following sentence in Conclusion section from (Line 366 - 371)

“Nutritional intervention with whey protein and vitamin D in patients with subacute post-stroke hemiparesis led to suppression of fat infiltration into the muscle. Increased intramuscular fat induces insulin resistance, and suppressing intramuscular fat accumulation is expected to reduce the risk of metabolic complications in stroke survivors.”

To

“Although no significant change was observed in muscle mass and strength, Nutritional intervention with whey protein and vitamin D in patients with subacute post-stroke hemiparesis led to suppression of fat infiltration into the muscle.”

  1. Supplementing with vitamin D and whey protein combined does not allow us to understand which of the two supplements may have given the improvement in intramuscular fat observed in the study group.

Reply:

Thank you for the comment. As reviewer pointed out, in our study design using commercial product, it might be difficult to evaluate the effects of each nutrients individually.

However, in both whey protein and vitamin D, there is some evidence to improve lipid metabolism and reduce body fat. Therefore, both of these lipid metabolism-improving effects of whey protein and vitamin D may have been contributed to the reduced fat infiltration observed in the HP group.

To deepen the discussion, we have added the text (Line 271 – 296) with following References.

  1. Miller, P.E.; Alexander, D.D.; Perez, V. Effects of whey protein and resistance exercise on body composition: a meta-analysis of randomized controlled trials. Journal of the American College of Nutrition 2014, 33, 163-175, doi:10.1080/07315724.2013.875365.

  1. Chen, W.-C.; Huang, W.-C.; Chiu, C.-C.; Chang, Y.-K.; Huang, C.-C. Whey protein improves exercise performance and biochemical profiles in trained mice. Medicin & Science in Sports & Exercise 2014, 46, 1517-1524, doi:10.1249/MSS.0000000000000272.

  1. Tauriainen, E.; Storvik, M.; Finckenberg, P.; Merasto, S.; Martonen, E.; Pilvi, T.K.; Korpela, R.; Mervaala, E.M. Skeletal muscle gene expression profile is modified by dietary protein source and calcium during energy restriction. Journal of Nutrigenetics and Nutrigenomics 2011, 4, 49-62, doi:10.1159/000327132.

  1. Chanet, A.; Salles, J.; Guillet, C.; Giraudet, C.; Berry, A.; Patrac, V.; Domingues-Faria, C.; Tagliaferri, C.; Bouton, K.; Bertrand-Michel, J., et al. Vitamin D supplementation restores the blunted muscle protein synthesis response in deficient old rats through an impact on ectopic fat deposition. The Journal of Nutrtional Biochemistry 2017, 46, 30-38, doi:https://doi.org/10.1016/j.jnutbio.2017.02.024.

  1. Thomas, M.K.; Lloyd-Jones, D.M.; Thadhani, R.I.; Shaw, A.C.; Deraska, D.J.; Kitch, B.T.; Vamvakas, E.C.; Dick, I.M.; Prince, R.L.; Finkelstein, J.S. Hypovitaminosis D in Medical Inpatients. The New England Journal of Medicine 1998, 338, 777-783, doi:10.1056/NEJM199803193381201.

  1. Sun, Q.; Pan, A.; Hu, F.B.; Manson, J.E.; Rexrode, K.M. 25-Hydroxyvitamin D levels and the risk of stroke: a prospective study and meta-analysis. Stroke 2012, 43, 1470-1477, doi:10.1161/STROKEAHA.111.636910.

  1. Shab-Bidar, S.; Neyestani, T.R.; Djazayery, A.; Eshraghian, M.R.; Houshiarrad, A.; Gharavi, A.; Kalayi, A.; Shariatzadeh, N.; Zahedirad, M.; Khalaji, N., et al. Regular consumption of vitamin D-fortified yogurt drink (Doogh) improved endothelial biomarkers in subjects with type 2 diabetes: a randomized double-blind clinical trial. BMC Med 2011, 9, 125, doi:10.1186/1741-7015-9-125.

  1. Many nutritional data have not been evaluated. Dosage of pre-albumin, tranferrin, and complement factors would have been very useful in assessing whether supplementation may have had real utility in improving the body composition status of the subjects.

Reply:

Thank you for the comment. We agree that pre-albumin, transferrin, and complement factors are important and useful indicators of nutritional state. However, in long-term intervention study such as this study, albumin which has long plasma half-time might be more appropriate to assess the nutritional state. So, we evaluated serum albumin level as the nutritional state indicator. It is noted that pre-albumin was also evaluated in this study (Table 4).

  1. There are important differences between the two groups after the supplementation period. The study group had a major reduction in weight and TGs. How do the authors justify these differences with the placebo group?

Reply:

Although BMI and TG decreased significantly from baseline to endpoint in the HP group, there were no significant differences between the groups.  The hospital meal provided in this study was sufficient to maintain body weight because BMI in the Placebo group was not decreased. Therefore, we considered that decrease in BMI in HP group was the effect of whey protein and vitamin D supplementation.

As described above, both whey protein and vitamin D have been reported to reduce body weight, body fat and serum lipid profiles. As well as fat infiltration into muscle, whey protein and vitamin D may have been contribute to reduce BMI and TG in HP group.

To deepen the discussion, we have added the text (Line 271 – 296).

  1. It would be useful to evaluate the vitamin D blood data from the beginning of supplementation and the end.

Reply:

Thank you for the comment. We agree with the importance of vitamin D. Serum 25(OH)D has often been evaluated as indicator of nutritional status of vitamin D, because 25(OH)D is converted from vitamin D in the liver and has long half-time. ( J L Omdahl  et al. Am J Clin Nutr, 1982,  K Nakamura et al., Environ Health Prev Med

. 2000) Therefore, we evaluated serum level of 25(OH)D as indicator of nutritional status of vitamin D.

In this study, serum 25(OH)D was lower than normal level at baseline, and increased significantly in the HP group at endpoint.

We have added following sentences to discuss the serum 25(OH)D level in this study (Line 284 – 296).

“Similarly, vitamin D deficiency induces to increase body fat and fat infiltration into the muscle and to inhibit muscle protein synthesis in the aged animal [38]. Vitamin D insufficiency is commonly observed in the patient admitted in the hospital, including the patient ingesting vitamin D of the recommended daily amount [39]. Besides, lower 25(OH)D level are known to associate with the risk of stroke onset [40]. We also observed lower serum 25(OH)D in our patients. However in this study, serum 25(OH)D level increased to normal range during the intervention period in the HP group. Vitamin D supplementation with 40 μg (1600 IU) / day was sufficient to improve the serum 25(OH)D level in post-stroke rehabilitation patients. Benefical effect of vitamin D supplementation on serum fat profile such as TG was also reported in the human study [41]. Therefore, improvement of BMI, serum TG and fat infiltration into the muscle might have been contributed by the both of whey protein and vitamin D supplementation in this study.”

  1. Dietary assessment is almost absent. Knowing the type of diet followed and any differences in the quality and quantity of intake of macronutrients is essential.

Reply:

Thank you for the comment on this important point. As reviewer pointed out, the lack of detailed information about primary dietary intake is the limitation of this study (Line 358 - 360).

We provided hospital meal based on Harris-Benedict equation using ideal body weight with activity factor individually. Protein amount in hospital meal was set at 1.0 g / kg ideal body weight / day. Amount of whey protein supplementation was set as “1.2 - 1.5 g protein intake / kg ideal body weight / day” which is recommended dosage of the ESPEN Expert Group for the elderly having acute or chronic illness.

Therefore, we have added following sentence (Line 116- 123)

Hospital meal was individualized to implement each energy requirement calculated by Harris-Benedict equation [24] using ideal body weight based on body mass index (BMI) of 22.0 with stress factor 1.0 and activity factor 1.3. Protein content in the hospital meal was set at 1.0 g / kg ideal body weight / day. Amount of protein suppelemtation in the HP group was designed as sufficient amount for a 1.2 - 1.5 g / kg body weight recommended by the ESPEN Expert Group for the elderly having acute or chronic illness in total with hospital meal [8].”

Also, we have added following sentence (Line 272 - 276).

“In addition, significant decrease in TG and BMI from baseline were observed only in the HP group, meanwhile SMI were maintained from baseline. It was implied that the amount of meal calculated by the Harris-Benedict equation was enough to maintain body weight, because decreasing BMI from baseline was not observed in the Placebo group.”

  1. The discussion in which the possible benefit of intramuscular fat reduction on the risk of insulin resistance and diabetes is emphasized seems very forced and with little justification. The authors should revise the discussion and also emphasize the poor effects of supplementation considering also the costs.

Reply:

We greatly appreciate for reviewer for raising this important point. As mentioned above, we revised the contexture of Discussion section to emphasize the ineffectiveness in muscle mass and strength (Line 233 -270).

Also, we have changed the following sentence in Abstract from (Line 33 - 37)

“Supplementation with whey protein and vitamin D in post-stroke patients led to suppression of fat infiltration into the muscle, which was expected to reduce the future risk of metabolic complications. Long-term follow-up studies are needed.”

To

“Supplementation with whey protein and vitamin D in post-stroke rehabilitation patients led to suppression of fat infiltration into the muscle. Long-term follow-up studies are needed to verify whether this nutritional intervention provides substantial benefits for the prognosis of stroke survivors.”

Also, we have changed the following sentence in Conclusion section from (Line 366 - 371)

“Nutritional intervention with whey protein and vitamin D in patients with subacute post-stroke hemiparesis led to suppression of fat infiltration into the muscle. Increased intramuscular fat induces insulin resistance, and suppressing intramuscular fat accumulation is expected to reduce the risk of metabolic complications in stroke survivors.”

To

“Although no significant change was observed in muscle mass and strength, Nutritional intervention with whey protein and vitamin D in patients with subacute post-stroke hemiparesis led to suppression of fat infiltration into the muscle.”

Again, thank you for giving us the opportunity to strengthen our manuscript with your valuable comments and queries. We have worked hard to incorporate your feedback and hope that these revisions persuade you to accept our submission.

Sincerely,

Kaoru Honaga

Department of Rehabilitation Medicine, Juntendo University Graduate School of Medicine

2-1-1, Hongo, Bunkyo Wardy, Tokyo 113-8421, Japan

Tel:  +81-3-3813-3111, Fax: +81-3-5684-1861,

E-mail address: k.honaga.ps@juntendo.ac.jp

Round 2

Reviewer 1 Report

I think it is interesting to point out a quote from the time of stroke, and the relationship between age and sarcopenia.

Author Response

28 January 2022

Dear Reviewer

I wish in response to the decision on our manuscript, titled Investigation of the effect of nutritional supplementation with whey protein and vitamin D on muscle mass and muscle quality in subacute post-stroke rehabilitation patients: A randomized, single-blinded, placebo-controlled trial. (Manuscript ID.: nutrients-1532250).

We are grateful for your comment regarding our manuscript. We have carefully revised the manuscript according to the suggestion, addressing the comment to the best of our ability.

Indeed, the suggestion was very insightful and useful in revising this manuscript. Along with the revised manuscript, please find attached a point-by-point reply to each of the reviewer comments.

Thank you for your continued consideration. We hope that our responses and revisions are satisfactory, and that the manuscript is now deemed to be suitable for publication in Nutrients. I look forward to hearing from you.

Reply to Reviewers comments

I think it is interesting to point out a quote from the time of stroke, and the relationship between age and sarcopenia.

Reply:

Thank you for this important suggestion. In line with Reviewers suggestion, we have added following sentence in Introduction section (Line 62);

‘The incidence of stroke increases with aging.’

Also, we have added following sentences in Discussion section (Line 314-319);

‘Intramuscular fat accumulation is also increases with aging [43]. Increased intramuscular fat accumulation is associate with muscle strength [44] and is known to be a risk of mobility restriction [45]. Also, reduced muscle quality has been included into the operational definition of sarcopenia on EWGSOP2 [5]. Therefore, decreased intramuscular fat infiltration observed in this study may suppress the progression of sarcopenia in post-stroke elderly patients by maintaining their muscle quality.’

Also, we have added following literatures as References;

  1. Delmonico, M.J.; Harris, T.B.; Visser, M.; Park, S.W.; Conroy, M.B.; Velasquez-Mieyer, P.; Bodreau, R.; Manini, T.M.; Nevitt, M.; Newman, A.B.:, and Bret H Goodpaster, B.H. Longitudinal study of muscle strength, quality, and adipose tissue infiltration. The American journal of clinical nutrition 2009, 90, 1579-1585, doi:10.3945/ajcn.2009.28047.
  2. Goodpaster, B.H.; Carlson, C.L.; Visser, M; Kelley, D.E.; Scherzinger, A.; Harris, T.B.; Stamm, E.; Newman, A.B. Attenuation of skeletal muscle and strength in the elderly: The Health ABC Study. Journal of Applied Physiology 2001, 90, 2157-2165. doi: 10.1152/jappl.2001.90.6.2157.
  3. Visser, M.; Goodpaster, B.H.; Kritchevsky, S.B.; Newman, A.B.; Nevitt, M.; Rubin, S.M.; Simonsick, E.M.; Harris, T.B. Muscle mass, muscle strength, and muscle fat infiltration as predictors of incident mobility limitations in well-functioning older persons. Journal of Gerontology: Medical Science 2005, 60, 324-333. doi: 10.1093/gerona/60.3.324. ‘

Again, thank you for giving us the opportunity to strengthen our manuscript with your valuable comments and queries. We have worked hard to incorporate your feedback and hope that these revisions persuade you to accept our submission.

Sincerely,

Kaoru Honaga

Department of Rehabilitation Medicine, Juntendo University Graduate School of Medicine

2-1-1, Hongo, Bunkyo Wardy, Tokyo 113-8421, Japan

Tel:  +81-3-3813-3111, Fax: +81-3-5684-1861,

E-mail address: k.honaga.ps@juntendo.ac.jp

Reviewer 2 Report

The authors have made some changes to the paper. 
Three major problems remain.
1. It is clear that the differences between the two study groups are very small in relation to outcomes. From the title of the paper it should be emphasised that integration is largely ineffective in bringing about substantial improvements in the health of the subjects.
2. Without a detailed assessment of the subjects' diets it is not possible to reach objective conclusions on the effectiveness of supplementation.
3. Nutritional Supplementation with Whey Protein and  Vitamin D together does not allow to determine whether it is the vitamin D or the whey protein  or both that is effective

Author Response

26 January 2022

Dear Reviewer

I wish in response to the decision on our manuscript, titled ‘Nutritional supplementation with whey protein and vitamin D in patients undergoing subacute post-stroke rehabilitation suppresses fat infiltration into the muscle: A randomized, single-blinded, placebo-controlled trial’ (Manuscript ID.: nutrients-1532250).

We are grateful for your comments regarding our manuscript. We have carefully revised the manuscript according to these suggestions, addressing each comment to the best of our ability.

Indeed, the suggestions were very insightful and useful in revising this manuscript. Along with the revised manuscript, please find attached a point-by-point reply to each of the reviewer comments.

Thank you for your continued consideration. We hope that our responses and revisions are satisfactory, and that the manuscript is now deemed to be suitable for publication in Nutrients. I look forward to hearing from you.

Reply to Reviewers comments

  1. It is clear that the differences between the two study groups are very small in relation to outcomes. From the title of the paper it should be emphasised that integration is largely ineffective in bringing about substantial improvements in the health of the subjects.

Reply:

Thank you for the comment. We agree Reviewers opinion that we should not emphasise small positive effect alone in the title. Therefore, we have changed the title of our manuscript from;

‘Nutritional supplementation with whey protein and vitamin D in patients undergoing subacute post-stroke rehabilitation suppresses fat infiltration into the muscle: A randomized, single-blinded, placebo-controlled trial.’

To

‘Investigation of the effect of nutritional supplementation with whey protein and vitamin D on muscle mass and muscle quality in subacute post-stroke rehabilitation patients: A randomized, single-blinded, placebo-controlled trial.’

  1. Without a detailed assessment of the subjects' diets it is not possible to reach objective conclusions on the effectiveness of supplementation.

Reply:

Certainly, as Reviewer pointed out, it is important to assess the diets of the subject on the evaluation of the effects of nutritional supplementation. Subjects in this study were inpatients in the same facility and consumed same hospital meal which was organized by hospital dietitian. As shown Line 120 - 123, provided energy contents were calculated individually by Harris-Benedict equation using ideal body weight and protein contents in hospital meal were set at 1.0 g / kg ideal body weight. Also, as shown in Table 2, consumption rates of the hospital meal were relatively high (90 – 96 %) and no difference between the groups. Therefore, we considered that consumed nutrients from their hospital diets were not different between groups.

However, micronutrients were not identified in this study, we have mentioned this point in Discussion section as the limitation.

We have changed following sentences from (Line 362 - 371)

‘The lack of detailed information about primary dietary intake might be one of the limitations of this study. The effects of nutritional supplementation should be affected by the basic nutritional status of the subjects [19]. However, the subjects in this study consumed hospital meal at the same facility. The amount of energy and protein in the hospital meal were set based on ideal body weight individually. Therefore, it is considered that there were no large differences in dietary intake between the groups.’

To

‘There are several limitations in this study. First, we have no detailed information about the nutritional intakes from the daily meals. The subjects in this study consumed hospital meal at the same facility. The amount of energy and protein in the hospital meal were set based on ideal body weight individually. It is considered that there were no large differences in the macronutrients intakes between the groups. However, the amounts of micronutrients could not identify, these differences might have been affected the results.’

  1. Nutritional Supplementation with Whey Protein and Vitamin D together does not allow to determine whether it is the vitamin D or the whey protein or both that is effective.

Reply:

Thank you for the comment. We speculate that the effect observed in this study might have been contributed by the both of whey protein and vitamin D (Line 298 - 300). However, we agree that it will be desirable to evaluate the effect of whey protein and vitamin D individually, as a nutritional components.

We have added the following sentence in Discussion section as the limitation of our study (Line 371 - 374).

‘Second, we used commercially available products to evaluate the effect of whey protein and vitamin D on the muscle. This study design could not allow us to evaluate the contribution of each nutrients individually. Further study will be needed on each nutrients individually.’

Again, thank you for giving us the opportunity to strengthen our manuscript with your valuable comments and queries. We have worked hard to incorporate your feedback and hope that these revisions persuade you to accept our submission.

Sincerely,

Kaoru Honaga

Department of Rehabilitation Medicine, Juntendo University Graduate School of Medicine

2-1-1, Hongo, Bunkyo Wardy, Tokyo 113-8421, Japan

Tel:  +81-3-3813-3111, Fax: +81-3-5684-1861,

E-mail address: k.honaga.ps@juntendo.ac.jp
